# Polyphenolic Contents, Free Radical Scavenging and Cholinesterase Inhibitory Activities of *Dalbergiella welwitschii* Leaf Extracts

**DOI:** 10.3390/plants11152066

**Published:** 2022-08-08

**Authors:** Tabisa Diniso, Jerry Adeyemi, Ayodeji Oriola, Taiwo Elufioye, Mavuto Gondwe, Adebola Oyedeji

**Affiliations:** 1Department of Chemical and Physical Sciences, Walter Sisulu University, Mthatha 5099, South Africa; 2Department of Human Biology, Faculty of Health Sciences, Walter Sisulu University, Mthatha 5099, South Africa

**Keywords:** *Dalbergiella welwitschii*, neurodegenerative disorders, polyphenolic contents, free radical scavengers, cholinesterase inhibitors

## Abstract

A decoction of *Dalbergiella welwitschii* leaves has been used ethnomedicinally for the treatment of mental illness and inflammatory diseases amongst other diseases. In this study, the leaf methanol extract of *D. welwitschii* and its partition fractions: n-hexane, ethyl acetate and aqueous, were tested and evaluated for their polyphenolic contents, free radical scavenging and cholinesterase inhibitory activities. The total phenolic (TPC), flavonoid (TFC) and proanthocyanidin (TPA) contents were determined using standard colorimetric methods. The anti-radical activity of the extracts against the 2,2-diphenyl-1-picrylhydrazyl (DPPH), ferric ion and nitric oxide (NO) radicals as well as their effects on lipid peroxidation were monitored spectrophotometrically. The cholinesterase enzyme (AChE and BuChE) inhibitions by the extracts were determined by a modified method of Ellman. The result showed a concentration-dependent increase in inhibition of the free radicals and the cholinesterase enzymes, except for that of lipid peroxidation. The ethyl acetate (EtOAc) fraction exhibited the highest polyphenolic contents among the fractions, with a TPC of 1.08 mgGAE/g, TFC of 0.38 mgQuE/g and TPA of 0.21 mgGAE/g. It also demonstrated the highest free radical scavenging activities with 72.63% and 65.43% inhibitions of DPPH and NO, respectively. The EtOAc fraction inhibited AChE and BuChE enzymes with IC_50_ values of 0.94 and 8.49 mg/mL, respectively. Our findings show that the plant may have polyphenol contents, in particular in the methanol extract and EtOAc fraction. These extracts showed considerable free radical scavenging and cholinesterase inhibitory properties. Thus, the observed bioactivities may serve as a justification for its folkloric use as a remedy for mental illness. The study also provides relevant information that could help in the search for lead cholinesterase inhibitors from medicinal plants that can be exploited against neurodegenerative disorders.

## 1. Introduction

Neurodegenerative disorders (NDs) are a heterogeneous group of diseases that are characterized by the progressive degeneration of the structure and function of the brain, particularly the central- and/or peripheral nervous systems [1]. Common NDs include Alzheimer’s disease (AD), Parkinson’s disease (PD), Huntington disease (HD) and amyotrophic lateral sclerosis (ALS) [2]. These neurological conditions are often an irreversible and dynamic processes associated with age and characterized by the progressive and irreversible loss of cognitive abilities, memory loss, emotional dysfunction and changes in behaviour [3,4].

They usually occur due to the reduced levels of the neurotransmitter acetylcholine in the brain [5]. Several factors, including aging and some pathological conditions, such as cholinergic deficit, oxidative stress, impaired mitochondrial function and neuroinflammation have been associated with NDs; thus, their management involves addressing one or more of the associated conditions [6]. Currently, there are no known cures for NDs, and there is no way to slow down the damages it causes to the nerve in the brain, due to its complexity and not-fully-understood etiopathology [7].

Nevertheless, cholinesterase inhibitory drugs, such as tacrine, donepezil and rivastigmine, appear to help maintain mental coordination and slow down memory loss for a while. These inhibitors act by inhibiting acetylcholinesterase and increasing the level of acetylcholine in the synaptic junction, thereby, improving cholinergic neurotransmission [8]. However, they do not stop the progression of the disease and have also been associated with adverse side effects [7]. The adverse effects have thus led to the need for more effective and less toxic cholinesterase inhibitory agents from medicinal plants.

Furthermore, proofs have emerged implicating free radical-induced oxidative damages in the pathogenesis of Alzheimer’s disease [9,10]. Free radicals are reactive oxygen species (ROS) that attack and damage lipids, proteins and DNA. The brain is susceptible to oxidative damage because of its high content of readily oxidized fatty acids, high use of oxygen and low levels of antioxidants [10]. Antioxidants are chemical substances that terminate the free radicals generated within the biological system by scavenging the reactive species-causing oxidative injuries, thus, reducing the risk of NDs, such as AD and other diseases [11].

Many natural products from diverse plants have been identified as useful materials for the treatment of many NDs. One of such plant is *Spondias mombin*, used in traditional medicine for the management of memory loss. This plant has been reported to contain cholinesterase inhibitory compounds, such as betulin, campesterol and 3,7,11,15-tetramethyl-2-hexadecen-1-ol [12].

Plants in the genus *Dalbergia* have also been acclaimed to be used locally for the management of mental illness, inflammatory diseases amongst other diseases [13]. Compounds, such as isoflavones, isoflavanones, neoflavones, sterols, anthraquinones, cinnamyl esters, furans and triterpenes, have been identified in this genus [14,15]. Some species in this genus, such as *D. sissoo*, *D. paniculata* and *D. odorifera,* have shown considerable antioxidant and cholinesterase inhibitory properties [16,17,18]. However, one of the popular medicinal species, *D. welwitschii* is yet to be fully exploited; hence, our report.

*Dalbergiella welwitschii* (Baker) Baker f. (Fabaceae), commonly known as the West African blackwood is an unthreatened perennial shrub or climber commonly found in the dry deciduous, fringing forest, riverine forests and abandoned farmland, particularly from Guinea through Cameroun to Angola [16]. As with many plants in the family Fabaceae, it can be grown through its seeds [19,20]. However, *D. welwitschii* is mostly sourced from the wild for its medicinal values and for economic purposes, such as for the making wooden toys, drum hoops and other household items [21]. A decoction of the leaves of the plant is mostly used for the management of mental illness and inflammatory conditions, such as arthritis, rheumatism, stomach troubles and bronchial ailments [16,18,22]. The plant has been reported to possess considerable antioxidant [23], anti-inflammatory [24] and anthelmintic [25] activities.

We report the polyphenolic contents, free radical scavenging and cholinesterase inhibitory activities of the leaf methanol extract of *D. welwitschii* and its partition fractions. This is with a view to identify the most active fraction and potentiate its biological activities towards the management of NDs.

## 2. Results and Discussion

### 2.1. Polyphenolic Contents

The polyphenol contents of the plant extract and its partition fractions were measured in terms of their total phenolics, total flavonoids and total proanthocyanidins. These phytochemical contents are known to play an important role in the biological properties of extracts, vis a viz their free radical scavenging and cholinesterase inhibitory activities [26]. Our study showed that the ethyl acetate fraction exhibited the highest total phenolic contents (TPC) of 1.08 ± 0.01 mgGAE/g, while the lowest phenolic content was demonstrated by the aqueous fraction (Figure 1).

The methanol (mother) extract had 0.57 ± 0.01 mgQuE/g total flavonoid content (TFC) (Figure 2). Upon fractionation, the highest TFC was recorded in the ethyl acetate fraction with 0.38 ± 0.01 mgQuE/g, while the aqueous fraction had the least TFC with 0.09 ± 0.01 mgQuE/g. The same scenario was observed in the total proanthocyanidin (TPA) experiment, where the ethyl acetate fraction demonstrated the highest TPA among the fractions with 0.21 ± 0.01 mgGAE/g (Figure 3).

These findings are in consonance with the report of Lakshmi et al. [27], where the ethyl acetate extract of *D. sissoo* gave a higher absorbance values than its mother ethanol extract and consequently higher TPC. The ethyl acetate fraction of *D. odorifera* seed was reported to show the highest phenolic (563.2 mg/g) and flavonoid (350.3 mg/g) contents amongst four tested solvent fractions, according to Lianhe et al. [28]. Proanthocyanidin dimers were identified from the stem bark ethyl acetate component of *D. monetari* [29]. Flavonoids were also reported to account for a large part of the polyphenolic contents of the ethyl acetate and methanol extract of *D. odorifera* seed [28].

Thus, our study has shown that the organic (methanol) extract, and particularly the ethyl acetate fractions of *D. welwitschii* leaves, have high phenolic, flavonoid and proanthocyanidin contents, unlike the aqueous fraction with low TPC, TFC and TPA. These remarkable difference in the polyphenolic contents of *D. welwitschii* across the organic and aqueous divide could be ascribed to the high eluting strength of the organic solvents, particularly the ethyl acetate and methanol, for polyphenolic constituents and conversely, the low eluting strength of the aqueous solvent (water) for polyphenol constituents [30].

### 2.2. Free Radical Scavenging Activity

The free radical scavenging activities of the extract and fractions of *D. welwitschii* leaves as a measure of antioxidant property, were expressed in terms of their ability to reduce the green ferric ion to the blue ferrous ion, scavenge (decolorize) the purple DPPH radical and inhibit nitric oxide radicals, under colorimetric conditions. The ferric reducing antioxidant power (FRAP) of an extract is a measure of its ability to donate electrons; thus, it may serve as an important indicator of the antioxidant activity of natural products [31].

The FRAP analysis showed a concentration-dependent increase in the absorbance value from 125–1000 µg/mL and consequently a marked reduction of ferric ion radical to the harmless ferrous ion (Figure 4). The ethyl acetate fraction gave the best activity across all the test concentration, based on its highest absorbance values. Here, its activity was comparable to that of its mother extract at 250, 500 and 1000 µg/mL concentrations. This corroborates the report of Lianhe et al. [28] in which the ethyl acetate fraction of *D. odorifera* exhibited the highest reducing power (1230 µg/g) amongst the partition fractions.

The 2,2-diphenyl-1-picrylhydrazyl (DPPH) test is a fast and reliable colorimetric method, based on the reduction of DPPH radicals in the presence of a proton-donating substance, resulting in the formation of diamagnetic molecules [32]. The test samples showed a steady increase in the percentage inhibition of the DPPH radical from 49% to 73% with increase in concentration from 125–1000 µg/mL (Figure 5). At 1000 µg/mL, the standard antioxidant, L(+)-ascorbic acid gave the highest DPPH radical scavenging ability (87.19 ± 0.01%), closely followed by the ethyl acetate fraction at 72.63 ± 0.01% inhibition.

The capacity of the plant extract and fractions to inhibit the activity of inducible nitric oxide synthase (iNOS) by free radical scavenging was demonstrated in the nitric oxide (NO) radical inhibition test. The methanol extract gave the best activity, having demonstrated 79.32% inhibition of NO radical, while the ethyl acetate fraction demonstrated 65.43% NO inhibition (Figure 6). A previous study indicated the NO inhibitory potential of some flavonoid-rich exudates from the stem bark of *D. odorifera* within an IC_50_ range of 3.19–31.60 µM [33].

A non-enzymatic lipid peroxidation method was also used in this study where the inhibitory ability of the plant extracts for oxidative damage to lipids was monitored by effectively stopping chain reaction initiated by hydrogen abstraction or addition of an oxygen radical [34]. At 7.81 µg/mL, the methanol extract demonstrated over 90% inhibition of lipid peroxidation, which was the highest activity when compared to the partition fractions and positive controls (Figure 7). Therefore, the methanol extract of *D. welwitschii* may have acted as an exogenous (natural) free radical scavenger by reversibly increasing the antioxidant level and decreasing the physiological production of oxidants in the presence of operative endogenous antioxidant defences. Thus, the extract may inhibit the propagation of lipid radicals and ultimately prevent the destruction of membrane lipids [35].

### 2.3. Cholinesterase Inhibitory Activity

The cholinesterase inhibitory activity of the plant extract and fractions were evaluated regarding their ability to block the normal breakdown of acetylcholine to prevent neurodegeneration. It is noteworthy that acetylcholine is the main neurotransmitter found in the body, and it has mental functions in both the peripheral nervous system and the central nervous system [36].

The results showed a steady increase in the percentage inhibition of acetycholinesterase (AChE) and butyrylcholinesterase (BuChE) enzymes from 0.16 to 5.00 mg/mL (Figure 8 and Figure 9). The standard drugs, donezepil and eserine gave the highest AChE inhibition (≥50%) at 1.25 mg/mL. This was followed by the ethyl acetate fraction with 34% inhibition at the same concentration. On the other hand, the methanol extract showed the highest BuChE inhibition (≥44%) among the plant samples at 5 mg/mL. The final cholinesterase activity was presented in terms of their IC_50_ values (Table 1).

Here, the ethyl acetate fraction was the most active among the partition fraction with IC_50_ values of 0.94 and 8.48 mg/mL in the AChE and BuChE analysis, respectively. A report has shown that the leaf extract of *D. sissoo*, a closely related species, ameliorates the effect of Aβ (1-42) on cognitive functions and attenuates oxidative stress and neuroinflammation at 300 and 500 mg/kg body weight in male Wistar rats [18]. Therefore, our study demonstrated that the polyphenolic-rich ethyl acetate fraction of *D. welwitschii* exhibited cholinesterase inhibitory activity by inhibiting inducible iNOS and lipid peroxidation, reducing ferric ion free radical and the presence of a rich proton-donating group.

Evidence has emerged about the significant antioxidant and neuroprotective properties of certain flavonoids and other polyphenolic compounds [37]. Proanthocyanidins are known to play a key defensive role of the body against tissue damage and to improve blood distribution to the brain and other part of the body by reinforcing the blood vessels [38]. Therefore, it may not be far-fetched that the observed free radical scavenging and cholinesterase inhibitory activities of the plants could partly be due to its polyphenol-rich ethyl acetate fraction.

In summary, our findings show that the phenolic, flavonoid and proanthocyanidin contents (polyphenols) of *D. welwitschii* leaves may reside in the methanol extract and the ethyl acetate fraction. These two components also showed considerable free radical scavenging and cholinesterase inhibitory properties. As such, the plant may be potentially useful in the management of neurodegenerative conditions.

## 3. Materials and Methods

### 3.1. Chemicals, Reagents, Solvents and Equipment

All chemicals, reagents and solvents were of analytical grade, purchased from Shalom Laboratory Suppliers, South Africa, an outlet for Sigma–Aldrich Chemical Co. (St Louis, MO, USA). A Bio-Rad 680 microplate reader (Bio-Rad Technologies, Hercules, CA, USA). Heidolph Laborota 400 Rotovap. rotary evaporator (Schwabach, Germany).

### 3.2. Plant Material

*Dalbergiella welwitschii* was collected in the month of May 2018 at an abandoned farmland along Jericho area, Ibadan, Oyo state, Nigeria (DMS Latitude: 7°22′36.2496″ N; DMS Longitude: 3°56′23.2296″ E). The plant was identified and authenticated by Mr T.K. Odewo, a taxonomist at Forest Research Institute of Nigeria (FRIN) Ibadan, Nigeria. Voucher specimen, FHI 112217 was deposited at the Department of Pharmacognosy herbarium, University of Ibadan, Nigeria.

### 3.3. Extraction and Fractionation

Fresh leaves of *Dalbergiella welwitschii* were air-dried under the shade in a well-ventilated area. The dried leaves were milled into powder. The powdered plant sample (1.04 kg) was macerated in 5 L of methanol for 72 h with intermittent shaking. It was filtered and concentrated in vacuo using rotary evaporator set at 40 °C and 100 rpm. The methanol extract (113.12 g) was partitioned into n-hexane, ethyl acetate and distilled water to obtain the respective fractions, according to the standard method previously described by Otsuka [39].

### 3.4. Determination of Polyphenolic Contents

The polyphenolic contents of the plant extract and fractions were determined based on the use of standard assay procedures as described below.

#### 3.4.1. The Total Phenolic Content

The total phenolic content (TPC) of each plant sample was determined by a modified colorimetric Folin–Ciocalteu method [40]. About 2.5 mL of 10% Folin–Ciocalteu reagent and 2 mL of Na_2_CO_3_ (2% *w*/*v*) were added to 0.5 mL of 1 mg/mL of each plant sample in triplicates. The resulting mixture was incubated at 45 °C for 15 min, after which the absorbance was measured at a 765 nm wavelength on a microplate reader. A standard calibration curve was prepared to estimate the TPC, using Gallic acid as an internal standard at a concentration range of 0.031–1.000 mg/mL.
(1)Y=0.0451x+0.3921

The TPC was expressed as milligram of Gallic acid equivalent per gram (mgGAE/g) of each plant sample.

#### 3.4.2. The Total Flavonoid Content

The total flavonoid content (TFC) of each plant sample was determined using a modified colorimetric method of Ordonez et al. [41]. Here, 1 mL each of the test samples (1 mg/mL) was mixed with 3 mL of methanol, 0.2 mL of 10% aluminium chloride, 0.2 mL of 1 M potassium acetate and 5.6 mL of distilled water, all in triplicates. The reaction mixture was kept at room temperature for 30 min. It was then transferred to a microplate reader set at 415 nm wavelength for absorbance measurement. A standard calibration curve was prepared to estimate the TFC, using Quercetin as the internal reference at a concentration range of 0.031–1.000 mg/mL.
(2)Y=0.0031x+0.2110

The total flavonoid content was expressed as milligram of Quercetin equivalent per gram (mgQuE/g) of each test sample.

#### 3.4.3. The total Proanthocyanidin Content

The total proanthocyanidin content (TPA) of each plant sample was determined based on the procedure previously described by Mbaebie et al. [42]. This involves a mixture of vanillin in methanol (4% *v*/*v*) and 1.5 mL of hydrochloric acid added to 0.5 mL (1 mg/mL) of each test sample in triplicates. The resulting mixture was vortexed and allowed to stand at room temperature (≈25 °C) for 15 min. It was then read spectrophotometrically on a microplate reader set at 500 nm wavelength on a microplate reader. A standard curve was prepared to estimate the TPA, using Gallic acid as the reference compound at a concentration range of 0.031–1.000 mg/mL TPA was expressed as milligram of Gallic acid equivalent per gram (mgGAE/g) of test sample, based on the calibration curve of gallic acid at varying concentrations (1.0, 0.5, 0.25, 0.125, 0.063 and 0.031 mg/mL).
(3)Y=0.0019x+0.3545

### 3.5. Free Radical Scavenging Tests

The ability of the plant samples to scavenge free radicals, such as Fe^2+^, DPPH, NO and lipid peroxidation, was determined based on their spectrophotometric measurements as described in the following assay procedures.

#### 3.5.1. Determination of Ferric Reducing Antioxidant Power

The ferric reducing antioxidant power (FRAP) of each test sample was evaluated according to the method of Kumar and Helmaltha [43]. Here, 1.0 mL of each of the test samples in methanol (1.0–0.125 mg/mL) was added to the mixture containing 2.5 mL of phosphate buffer (0.2 M pH 6.6) and 2.5 mL of potassium ferricyanide [K_3_Fe (CN)_6_] (1% *w*/*v*) in triplicates. The resulting mixture was incubated at 50°C for 20 min, followed by the addition of 2.5 mL of trichloroacetic acid (TCA) (10% *w*/*v*).

It was then centrifuged at 300 rpm for 10 min. About 2.5 mL of the upper layer of the solution was mixed with 2.5 mL of distilled water and 0.5 mL of FeCl_3_ (0.1% *w*/*v*) in triplicates. It was transferred to a microplate reader and the absorbance was read at 655 nm wavelength. *L*-Ascorbic acid and 2.6-Di-tert-butyl-4-methyl phenol [(CH_3_)_3_C]_2_ C_6_H_2_(CH_3_) OH] (DDM) were used as the positive controls. Higher reducing power of each test sample was indicated by higher absorbance values.

#### 3.5.2. 2,2-Diphenyl-1-picrylhydrazyl Radical Scavenging Assay

The free radical scavenging activity of each test sample was carried out using a 2,2-diphenyl-1-picrylhydrazyl (DPPH) radical scavenging assay method, previously described by Shen et al. [44]. A solution of 0.135 mg/mL DPPH in methanol was prepared, and 1.0 mL of the DPPH solution was mixed with 1.0 mL of each of the test samples at varying concentrations (1.000–0.125 mg/mL) in triplicates. The reaction mixture was thoroughly mixed and kept in a dark room for 30 min. The absorbance was measured at 490 nm wavelength on a microplate reader. *L*-Ascorbic acid and DDM were used as the positive controls. The DPPH radical scavenging activity of the test samples was determined based on the equation:(4)$$\%\;\mathrm{DPPH}\;\mathrm{scavenging}\;\mathrm{activity}=\left[\frac{A0-A1}{A0}\right]*100$$
where *A*0 = absorbance of the control and *A*1 = absorbance of samples.

#### 3.5.3. Nitric Oxide Inhibitory Assay

The method of Ebrahimzadeh et al. [45] was used to determine the nitric oxide (NO) inhibitory activity of the extract and fractions. This involves the dissolution of about 2 mL of 10 mM sodium nitroprusside in 0.5 mL phosphate buffer saline (pH 7.4) was mixed with 0.5 mL of test samples at varying concentrations (1.0–0.125 mg/mL) in triplicates.

The mixture was incubated at 25 °C for 150 min, and an initial absorbance (*A*0) was taken at 540 nm. Thereafter, 0.5 mL of the incubated mixture was mixed with 1 mL of a sulfanilic acid reagent and 1 mL of naphthylethylenediamine dichloride (0.1% *w*/*v*). It was incubated at room temperature for 30 min before another absorbance (*A*1) was taken at 540 nm. *L-*Ascorbic acid and DDM were used as the positive controls. The percentage NO radical scavenging activity of the test samples was calculated using the following formula:(5)$$\%\;\mathrm{Inhibition}\;\mathrm{of}\;\mathrm{NO}=\left[\frac{A0-A1}{A0}\right]*100\;$$
here *A*0 = absorbance reaction and *A*1 = absorbance after the reaction.

#### 3.5.4. Lipid Peroxidation Assay

Lipid peroxidation assay was performed by measuring the lipid peroxide formed, using egg yolk homogenates as lipid-rich media according to the modified method of Banerjee et al. [46]. Here, an egg homogenate (0.5 mL of 10% *v*/*v*) and 0.1 mL of each test sample (1.0–0.78 mg/mL) were added to a test tube and made up to 1 mL with distilled water, in triplicates. A 0.05 mL of FeSO_4_ (0.07 M) was added to induce lipid peroxidation and the mixture was incubated for 30 min. Then, 1.5 mL of 3.5 M acetic acid, 1.5 mL of 0.06 M TBA in 0.04 M sodium dodecyl sulphate and 0.05 mL of 1.2 M TCA were added.

The resulting mixture was vortexed and heated at 95 °C for 60 min. To eliminate non-MDA interference, another set of samples was treated in the same way, incubated without TBA, to subtract the absorbance of each test sample. After cooling, 5 mL of butan-1-ol was added to each sample in a test tube and centrifuged for 10 min. The absorbance (A) of the organic layer was measured at 532 nm. *L*- Ascorbic acid was used as the positive control. Inhibition of lipid peroxidation (%) by the sample was calculated according to the formula:(6)$$\%\;\mathrm{Inhibition}=\left(1-\frac{E}{C}\right)*100$$
where,
(7)$$E=(A_{532}+TBA)-\left(A_{532}-TBA\right)$$

*C* is the absorbance value of the fully oxidized control.

### 3.6. Cholinesterase Inhibitory Tests

#### 3.6.1. Acetylcholinesterase (AChE) Inhibitory Assay

The AChE inhibitory activities of the extract and fractions were determined using standard colorimetric method as described by Ellman et al. [47]. This entails the use of a reaction mixture comprising 240 μL of buffer (50 mMTris-HCl, pH 8.0), 20 μL of varying concentrations of the extract/fractions (1, 0.5, 0.25, 0.125, 0.0625 and 0.03125 mg/mL) and 20 μL of the enzyme in triplicates. This reaction mixture was incubated for 30 min at 37 °C, after which 20 μL of 10 mM DTNB was added and the reaction started by the addition of 20 μL of the substrate (25 mM ATChI).

The rate of hydrolysis of the substrate was determined spectrophotometrically at 412 nm every 30 s for 4 min by measuring the change in absorbance per minute due to the formation of the yellow 5-thio-2-nitrobenzoate anion. A buffer solution was used as negative control, while Serine and Donepezil were used as positive controls. The percentage cholinesterase inhibition was calculated thus:(8)$$\%\;\mathrm{AChE}\;\mathrm{Inhibition}\;\left(I\right)=\frac{\left[Vo-Vi\right]}{Vo}*100$$
where: *I* (%) = Percentage inhibition, *Vi* = enzyme activity in the presence of extract, *Vo* = enzyme activity in the absence of extract.

The final activity was expressed in terms of the concentration that caused 50% inhibition of the AChE enzyme, referred to as the IC_50_ value, which was expressed as mean ± standard deviation of three replicate values.

#### 3.6.2. Butyrylcholinesterase (BuChE) Inhibitory Assay

The BuChE inhibitory activities of the extract and fractions were determined using standard colorimetric method as described by Ellman et al. [47]. Here, the reaction mixture consisted of 240 μL of buffer (50 mMTris-HCl, pH 8.0), 20 μL of varying concentrations of the extract/fractions (1, 0.5, 0.25, 0.125, 0.0625 and 0.03125 mg/mL) and 20 μL of the enzyme in triplicates. This reaction mixture was incubated for 30 min at 37 °C, after which 20 μL of 10 mM DTNB was added and the reaction started by the addition of 20 μL of the substrate (25 mM of BuTChI). The rate of hydrolysis of the substrate was determined spectrophotometrically at 412 nm every 30 s for 4 min by measuring the change in absorbance per minute due to the formation of the yellow 5-thio-2-nitrobenzoate anion. A buffer solution was used as negative control while Serine and Donepezil were used as positive controls. The percentage BuChE inhibition was calculated thus:(9)$$\%\;\mathrm{BuChE}\;\mathrm{Inhibition}\;\left(I\right)=\frac{\left[Vo-Vi\right]}{Vo}*100$$
where: *I* (%) = Percentage inhibition, *Vi* = enzyme activity in the presence of extract, *Vo* = enzyme activity in the absence of extract.

The final activity was expressed in terms of the concentration that caused 50% inhibition of the BuChE enzyme, referred to as the IC_50_ value, which was expressed as mean ± standard deviation of three replicate values.

### 3.7. Statistical Analysis

The results obtained were expressed as mean ± standard error of mean (SEM) using Microsoft Excel version 365 (Microsoft Corporation, Washington, DC, USA). The results were analyzed using One-way analysis of variance (ANOVA), followed by the Student-Newman-Keul’s as the post-hoc test on a GraphPad Prism 5 (GraphPad Software Inc., San Diego, CA, USA) The confidence level was set at *p* < 0.05, and this was considered as significant.

## 4. Conclusions

*D. welwitschii* leaves may have polyphenolic contents that are traceable to its methanol extract and ethyl acetate fraction. These extracts showed considerable levels of free radical scavenging and cholinesterase inhibitory properties. Thus, our findings may provide justification for the ethnomedicinal use of the plant as a remedy for mental illness. The study also provides relevant information, which may be helpful in the search for lead cholinesterase inhibitors from medicinal plants that can be exploited against neurodegenerative diseases.

## Figures and Tables

**Figure 1 plants-11-02066-f001:**
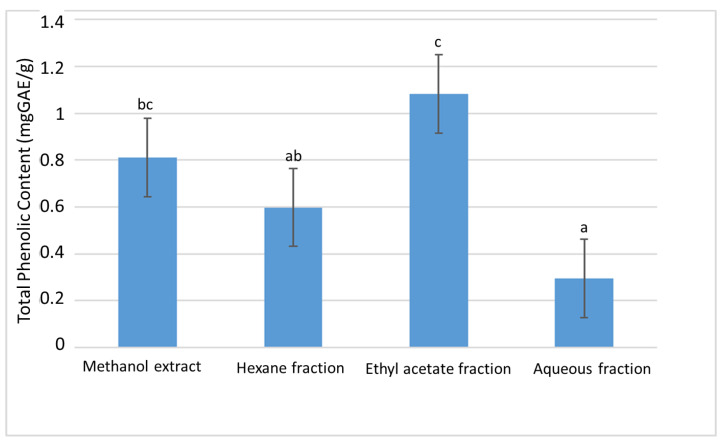
The total phenolic content (TPC) of the extract and fractions of *D. welwitschii* leaves. Footnote: Values with different superscript of alphabets within bars are significantly different (*p* < 0.05, One-way ANOVA, followed by the Student-Newman-Keuls post hoc test); and the total phenolic content expressed as milligram of gallic acid equivalent per gram of sample (mgGAE/g).

**Figure 2 plants-11-02066-f002:**
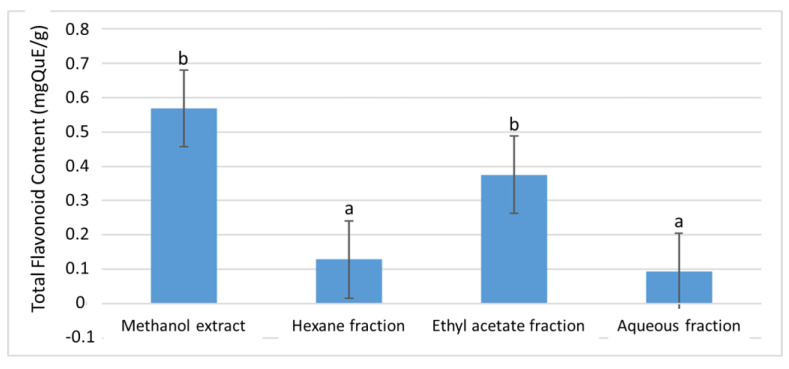
The total flavonoid content of the extract and fractions of *D. welwitschii* leaves. Footnote: Values with different superscript of alphabets within bars are significantly different (*p* < 0.05, One-way ANOVA, followed by the Student-Newman-Keuls post hoc test); and total flavonoid content expressed as milligram of Quercetin equivalent per gram of sample (mgQuE/g).

**Figure 3 plants-11-02066-f003:**
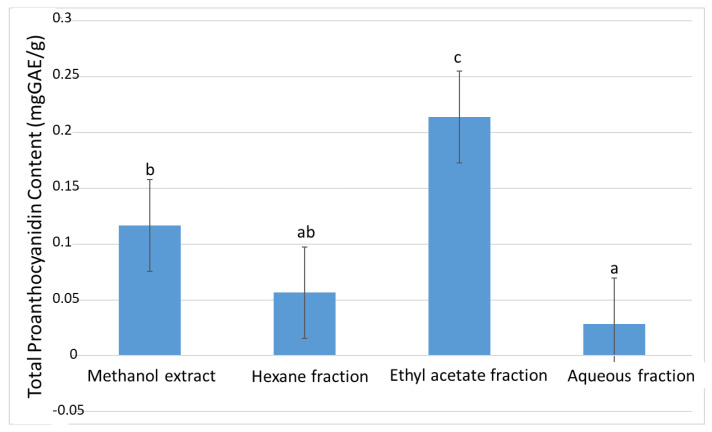
The total proanthocyanidin content of the extract and fractions of *D. welwitschii* leaves. Footnote: Values with different superscript of alphabets within bars are significantly different (*p* < 0.05. One-way ANOVA, followed by the Student-Newman-Keuls post hoc test).

**Figure 4 plants-11-02066-f004:**
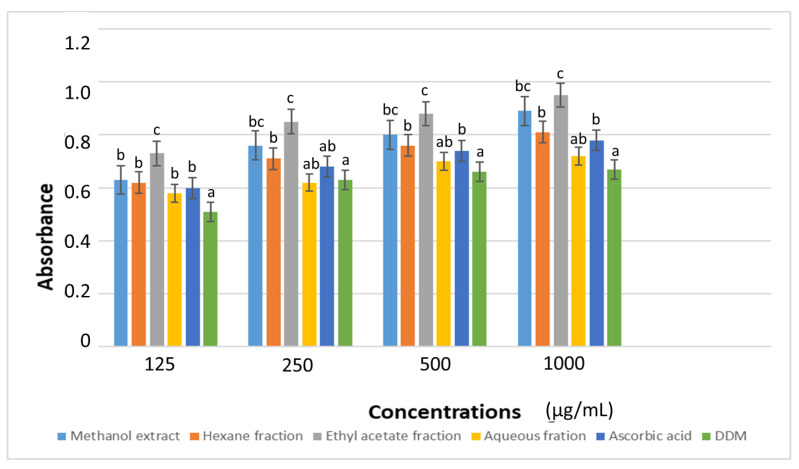
Ferric reducing antioxidant power of *D. welwitschii* leaf extract and fractions. Footnote: Values with different superscript of alphabets within bars are significantly different (*p* < 0.05, One-way ANOVA, followed by the Student-Newman-Keuls post hoc test). Positive control: 2,6-Di-tert-butyl-4-methyl-phenol (DDM).

**Figure 5 plants-11-02066-f005:**
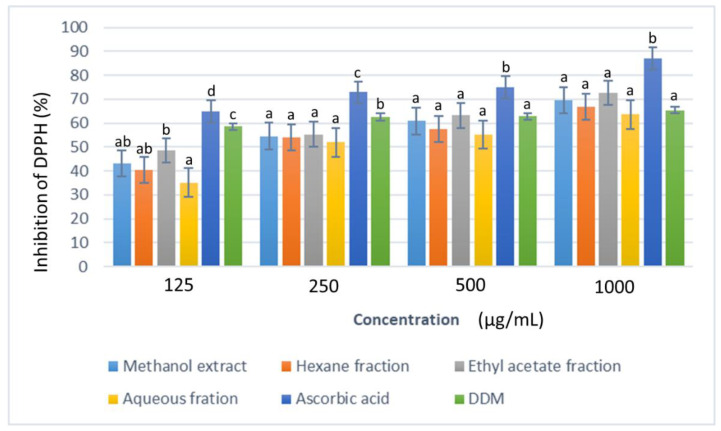
The DPPH radical scavenging activity of *D. welwitschii* leaf extract and fractions. Footnote: Values with different superscript of alphabets within bars are significantly different (*p* < 0.05, One-way ANOVA, followed by the Student-Newman-Keuls post hoc test). Positive control: 2,6-Di-tert-butyl-4-methyl-phenol (DDM).

**Figure 6 plants-11-02066-f006:**
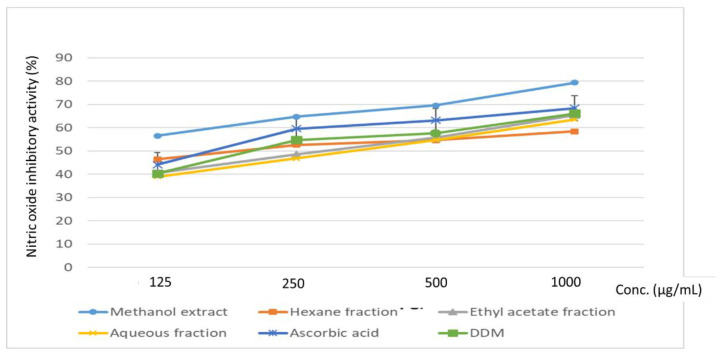
The nitric oxide scavenging activity of *D. welwitschii* leaf extract and fractions. Positive control: 2,6-Di-tert-butyl-4-methyl-phenol (DDM).

**Figure 7 plants-11-02066-f007:**
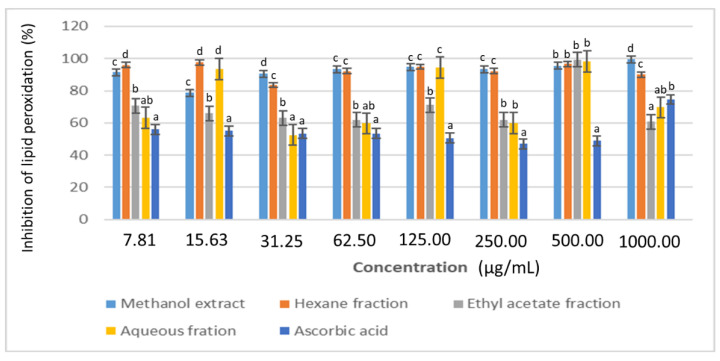
Effects of *D. welwitschii* leaf extract and fractions on non-enzymatic lipid peroxidation. Footnote: Values with different superscript of alphabets within bars are significantly different (*p* < 0.05, One-way ANOVA, followed by the Student-Newman-Keuls post hoc test).

**Figure 8 plants-11-02066-f008:**
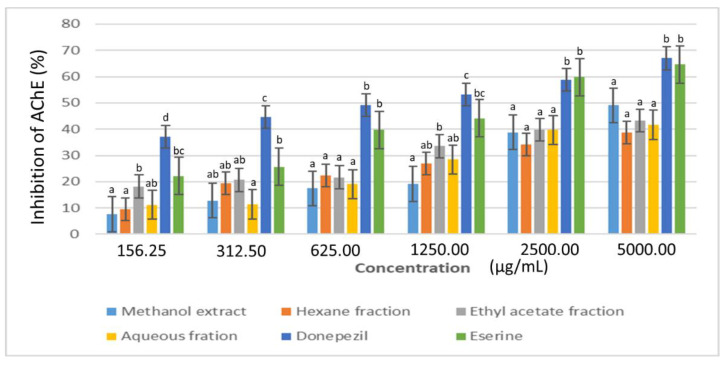
Acetylcholinesterase (AChE) inhibitory activity of the extract and fractions of *D. welwitschii* leaves. Footnote: Values with different superscript of alphabets within bars are significantly different (*p* < 0.05, One-way ANOVA, followed by the Student-Newman-Keuls post hoc test). Acetylcholinesterase enzyme (AChE).

**Figure 9 plants-11-02066-f009:**
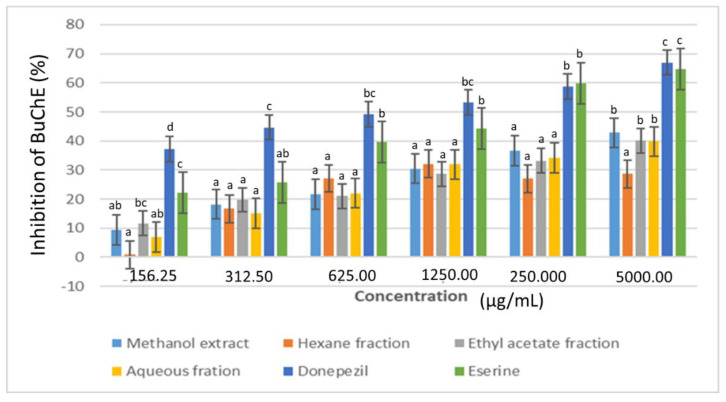
Butyrylcholinesterase (BuChE) inhibitory activity of the extract and fractions of *D. welwitschii* leaves. Footnote: Values with different superscript of alphabets within bars are significantly different (*p* < 0.05, One-way ANOVA, followed by the Student-Newman-Keuls post hoc test). Butyrylcholinesterase enzyme (BuChE).

**Table 1 plants-11-02066-t001:** The cholinesterase inhibitory activity of *Dalbergiella welwitschii* leaf.

Assay	IC_50_ ± SEM (mg/mL)					
	MeOH	Hexane	EtOAc	Aqueous	Eserine	Donepezil
AChE	0.45 ± 0.03 ^c^	5.27 ± 0.10 ^e^	0.94 ± 0.05 ^d^	7.43 ± 1.25 ^f^	0.002 ^b^	0.001 ^a^
BuChE	2.64 ± 0.05 ^c^	11.86 ± 1.25 ^e^	8.48 ± 2.25 ^d^	21.25 ± 1.50 ^f^	0.002 ^b^	0.001 ^a^

*n* = 3, data expressed as IC_50_ ± standard error of the mean (SEM); values with different alphabets in superscript are significantly different (*p* < 0.05, followed by one-way ANOVA and the Student–Newman–Keuls post hoc test); concentration that caused 50% inhibition of cholinesterase enzyme (IC_50_), acetylcholinesterase (AChE), butyrylcholinesterase (BuChE), extract (MeOH) and partition fractions (hexane, EtOAc and aqueous).

## Data Availability

Not applicable.

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
