# Peer review of "Polyphenolic Contents, Free Radical Scavenging and Cholinesterase Inhibitory Activities of Dalbergiella welwitschii Leaf Extracts"

_plants, 2022, doi:10.3390/plants11152066_

Round 1

Reviewer 1 Report

The evaluation of the manuscript entitled “Polyphenolic Con-tents, Free Radical Scavenging and Cholinesterase Inhibitory Activities of Dalbergiella welwitschii Leaf Extracts.” by Diniso et al. sent to Plants.

The manuscript requires some improvements and explanations before possible acceptance for publication.

Specific comments and suggestions:

Abstract: it contains too much methodological details. Please focus on results and conclusions.

Imbalances in levels of acetylcholine play a role in some neurological conditions. People who have Alzheimer's disease and Parkinson's disease tend to have low levels of acetylcholine. So, please in the first paragraph of the Introduction section mention the neurodegenerative disorders as the area and not only Alzheimer disease (AD).

Line 47: omit (de la Torre, 2004).

Figure 4: use FRAP Units (Fe2+ µg·mL−1)

Figures 8 and 9 etc.: no statistics?

In conclusion please state what phenolic fraction (not ethyl acetate fraction) is considered as most active towards parameters measured. It is possible to draw such conclusion taking into account your results?

Author Response

Response to Reviewer 1 Comments

Point 1: The manuscript requires some improvements and explanations before possible acceptance for publication.

Response 1: The manuscript has been improved substantially and more clarity given in the body.

Point 2: Abstract: it contains too much methodological details. Please focus on results and conclusions.

Response 2: The abstract methodology has been reduced. The major results are presented and the conclusion has been revised.

Point 3: Imbalances in levels of acetylcholine play a role in some neurological conditions. People who have Alzheimer's disease and Parkinson's disease tend to have low levels of acetylcholine. So, please in the first paragraph of the Introduction section mention the neurodegenerative disorders as the area and not only Alzheimer disease (AD).

Response 3: The introduction has been revised to put the work in proper perspective. The first paragraph has been revised to introduce neurodegenerative disorders (NDs) in a broad sense. More emphasis has now been placed on NDs as the main thrust on which Alzheimer's, Parkinson's and other neurological disorders can be found. This has generated 2 more citations [1] and [2] to the list of references.

Point 4: Line 47: omit (de la Torre, 2004).

Response 4: This has been revised accordingly

Point 5: Figure 4: use FRAP Units (Fe2+ µg·mL−1)

Response 5: The FRAP Units µg·mL−1 is now used Figure 4, in place of mg/mL

Point 6: Figures 8 and 9 etc.: no statistics?

Response 6: Figures 8, 9, and others, including table 1, have now been subjected to statistical analysis using one-way ANOVA, followed by Student-Newman-Keuls' post-hoc test, where p<0.05 was considered significantly different. A subsection marked "3.7" has been included in the manuscript to capture the method of result analysis.

Point 7: In conclusion please state what phenolic fraction (not ethyl acetate fraction) is considered as most active towards parameters measured. It is possible to draw such conclusion taking into account your results?

Response 7: Rather than describing the ethyl acetate fraction to be polyphenolic rich, the conclusion has now been revised that: "D. welwitschii leaves may have some polyphenolic contents, traceable to its methanol extract and ethyl acetate fraction. These extracts showed considerable free radical scavenging and cholinesterase inhibitory properties".

Reviewer 2 Report

This manuscript is important because provides relevant information on the polyphenolic contents, free radical scavenging and cholinesterase inhibitory activities of the leaf methanol extract of D. welwitschii and its partition fractions.

The provided results have a great practical application due to the fact that allow to identify the most active fraction in order to be further used towards the management of mental health-related problems.

The research is clearly designed and organized. The Introduction provides the reason for this research and an overview of research performed in this topic. The methods of analysis are well-presented, the results are discussed appropriately but the graphical representations can be improved. The cited references are relevant to this subject, being written in accordance with the requirements of the journal.

Recommended corrections:

- Please, revise the figures 1 – 3 and add the measurement units for total phenolic content (TPC), total flavonoid content (TFC) and total proanthocyanidin content (TPA).

- Also the figures 4 – 7, put the measurement units in the brackets: concentration (mg/ml) or (µg/ml), Inhibition of DPPH free radical (%), and so on.

- Please, make figure 6 clearer by moving down the legend with the methanolic extract and the fractions .

- Please, remove the line 114.

- Please, number the calculation formulas included in the Materials and Methods section.

- Statistical analysis of obtained data is missing. Please, include a section of statistical data processing to determine the significant differences in the evaluated bioactive properties (polyphenolic contents, free radical scavenging and cholinesterase inhibitory activities) between leaf extract and fractions. It could be used one-way analysis of variance (ANOVA) or other test in this regard.

- Please, improve the conclusions, making them more consistent, according to the obtained results.

Given that the content of this manuscript fits well with the purpose of the journal, I recommend its publication after performing the recommended corrections

Author Response

Response to Reviewer 2 Comments

Point 1: Please, revise the figures 1 – 3 and add the measurement units for total phenolic content (TPC), total flavonoid content (TFC) and total proanthocyanidin content (TPA).

Response 1: The figures 1 - 3 have been revised to indicate their measurement units for total phenolic content (TPC), total flavonoid content (TFC) and total proanthocyanidin content (TPA).

Point 2: Also the figures 4 – 7, put the measurement units in the brackets: concentration (mg/ml) or (µg/ml), Inhibition of DPPH free radical (%), and so on.

Response 2: The figures 4 - 7 have been revised to show the measurement units.

Point 3:  Please, make figure 6 clearer by moving down the legend with the methanolic extract and the fractions.

Response 3: Figure 6 has been re-formatted for more clarity.

Point 4: Please, remove the line 114.

Response: The line 114 "Figures" has been removed

Point 5: Please, number the calculation formulas included in the Materials and Methods section.

Response 5: The calculation formulas have now been numbered accordingly.

Point 6: Statistical analysis of obtained data is missing. Please, include a section of statistical data processing to determine the significant differences in the evaluated bioactive properties (polyphenolic contents, free radical scavenging and cholinesterase inhibitory activities) between leaf extract and fractions. It could be used one-way analysis of variance (ANOVA) or other test in this regard.

Response 6: A new subsection (3.7) has been included to describe the statistical analysis used in the study. The results presented in the figures and tables have now been subject to statistical analysis using One-way ANOVA, followed by the Student-Newman-Keul's post-hoc test for comparison. P-value was set at P<0.05, which was considered significant.

Point 7: - Please, improve the conclusions, making them more consistent, according to the obtained results.

Response 7: The conclusion has now been revised in consonance with the results obtained as thus: "D. welwitschii leaves may have some polyphenolic contents, traceable to its methanol extract and ethyl acetate fraction. These extracts showed considerable levels of free radical scavenging and cholinesterase inhibitory properties".

Reviewer 3 Report

The subject of the manuscript is very interesting and with possible practical implementation. The ethyl acetate fraction of D. welwitschii leaves is polyphenol-rich with considerable levelsof free radical scavenging and cholinesterase inhibitory propertiesI would recommend the publication of this work after listed issues are addressed.

General Comment:

This work is missing information how all methods were tested to ensure proper results are obtained. At least basic method transfer is needed to check specificity, range, repeatability etc. I believe such tests were performed but not described. This is especially important for spectrophotometric methods. A summary of these activities should be included. 

Specific Comments:

1. L.227: Plant material section: Indicate the applied agronomical practices: irrigation, fertilization, pruning, in addition temperatures during the harvest year (maximum, minimum, mean), the age of the plant. 2. L.247, L.258, L.266 I recommend that you describe the calibration curves. In accordance with the General Note, the reliability of the methods of analysis should be confirmed.

Author Response

Response to Reviewer 3 Comments

General Point: This work is missing information how all methods were tested to ensure proper results are obtained. At least basic method transfer is needed to check specificity, range, repeatability etc. I believe such tests were performed but not described. This is especially important for spectrophotometric methods. A summary of these activities should be included. 

Response to General Point: The authors have summarised and described the methods used in the study. The polyphenolic (TPC, TFC and TPA), free radical scavenging (DPPH, FRAP, nitric oxide and lipid peroxidation) and cholinesterase inhibition (AChE and BuChE) were carried out according to conventional/ standard assay procedures, which we have cited.

Point 1: L.227: Plant material section: Indicate the applied agronomical practices: irrigation, fertilization, pruning, in addition temperatures during the harvest year (maximum, minimum, mean), the age of the plant. 

Response 1: the requested information on agronomical practices and conditions pertains to how the plant is propagated. Please note, the study plant was not propagated for this investigation but was collected from the wild. Therefore, information on botanical description, geographical  distribution, plant habit, medicinal and economic values and other agronomy indicators are described in the introduction section, especially the paragraph on the botanical description and geographical distribution of the plant

Point 2: L.247, L.258, L.266 I recommend that you describe the calibration curves. In accordance with the General Note, the reliability of the methods of analysis should be confirmed.

Response 2: The calibration curves have been highlighted. The final results have now been statistically analysed by subjecting to one-way ANOVA and Student-Newman-Keul's post-hoc test. A 95% Confidence limit is tahrd as significant.